# Study of Preg-Robbing with Quicklime in Gold Cyanide Solutions Analyzed by Time-of-Flight Secondary Ion Mass Spectrometry

**Eber Manuel Garcia Rosales [1,*], Jesús Emilio Camporredondo Saucedo [1], Yuriy Kudriavtsev [2], Grover Johnny Mamani Maron [3], Fernando Rojas Venegas [3] and Laura Guadalupe Castruita Avila [4]**

[1] Facultad de Ciencias Químicas, Universidad Autónoma de Coahuila, Blvd. Venustiano Carranza 935, República, Saltillo 25280, Mexico; emiliocamporredondo@uadec.edu.mx

[2] Departamento Ingeniería Eléctrica—SEES, Cinvestav—IPN, Av. Instituto Politécnico Nacional 2508, San Pedro Zacatenco, Delegación Gustavo A. Madero, Mexico City 07360, Mexico

[3] Cal y Cemento Sur S.A., Carretera Juliaca Puno Km. 11, Juliaca 21107, Caracoto, Peru

[4] Facultad de Ingeniería Mecánica y Eléctrica, Universidad Autónoma de Coahuila, Barranquilla S/N, Guadalupe, Monclova 25720, Mexico; laura_castruita@uadec.edu.mx

\* Correspondence: ebergarcia@uadec.edu.mx; Tel.: +52-8662368275

**Abstract:** Preg-robbing is a phenomenon in which minerals retain gold, especially due to the presence of species like carbonaceous matter and silicates in the mineral. This study demonstrates the impact of quicklime, used to adjust the pH of a gold cyanidation solution, on the retention of gold contained in pregnant cyanidation solutions and sorption mechanisms. The retention capacity of four quicklime solutions was evaluated using proportions of 200 g of lime in 800 mL of solution and 10 g of lime in 500 mL of solution. The concentrations of the gold cyanide solutions were 10, 15, and 25 ppm. The insoluble lime residue in the acetic acid solution was separated and analyzed by XRD, FTIR, elemental carbon, and Raman spectroscopy techniques. SEM and TOF-SIMS were used to analyze the lime samples after exposure to the gold cyanide solution. The results show that retention was attributable to quicklime due to the effects of its carbon and silicate content, although chemisorption and physisorption mechanisms may also be responsible.

**Keywords:** preg-robbing; quicklime; TOF-SIMS

## 1. Introduction

The functions of quicklimes as pH regulators are essential in gold cyanidation. Different types of quicklime, like calcium oxide (CaO) and calcium hydroxide Ca(OH)$_2$, have various functions, such as preventing the loss of cyanide due to hydrolysis. Quicklime acts to neutralize acidic compounds in the mineral, such as ferrous and ferric salts, and aids in the sedimentation of fine mineral particles [1,2]. Quicklime is produced by thermolysis, that is, the calcination of limestone. Its main component is calcium oxide (CaO). The quality of quicklime depends on its physical properties, reactivity to water, and chemical composition [3]. The chemical equation of the thermolysis process that converts limestone into quicklime is as follows [4]:

$$CaCO_3 \text{ (s)} \longleftrightarrow CaO(s) + CO_2 \text{ (g)} \quad (\Delta H = +178 \text{ kJ/mol}) \quad (1000\,^{\circ}C) \tag{1}$$

The processes involved in producing and applying quicklime are relatively well-known and do not cause significant problems. However, so-called preg-robbing minerals, such as carlin-type minerals [5], can limit gold recovery by cyanidation due to the presence of natural carbonaceous matter and other species with the capacity to adsorb gold from cyanide solutions, a phenomenon that results in deficient recoveries. The capacity of carbonaceous matter to adsorb gold is widely recognized and documented [6], and

several theories on how adsorption of gold by carbonaceous matter occurs have been proposed. One holds that cyanide attaches to carbon as an ion pair and is adsorbed as a less soluble complex with potassium, sodium, or calcium [7]. Conversely, it has been suggested that aurocyanide adsorbs gold through electrostatic interactions with active sites on carbonaceous materials [8].

One of the most widely accepted theories revolves around the microstructure of carbonaceous matter and its degree of disorder [9]. It is well known that activated carbon has a disordered microcrystalline structure, and the random organization and oxidation during activation create a porous structure that explains its large surface area and notable adsorption capacity. It has been suggested that the anion $[Au(CN)_2]-^-$ is adsorbed with no change or chemical reaction parallel to the graphitic sheets of carbon, especially in sheet defects or edges. Various authors support this idea [10–12].

Helm et al. [13] used Raman spectroscopy to assess the physical nature of carbonaceous material and the degree of preg-robbing. In the study, graphitic carbon with high preg-robbing capacity had Raman spectra similar to those of activated carbon. In contrast, the spectra of graphitic carbon in minerals with low preg-robbing capacity aligned more closely with natural graphite.

Studies using novel characterization techniques and sufficient resolution have been conducted to detect trace elements in mineral phases. Dimov and Hart [14] applied the TOF-SIMS technique with electron microscopy to determine carbonaceous particles in samples of activated carbon from leaching processes. The sample of impregnated gold on the carbonaceous material showed the presence of both gold (Au) and the $Au(CN)_2$ compound. Amanya et al. [15] analyzed various carbonaceous materials, including wood chips, charcoal, barren carbon, and activated carbon. Their results showed an increasing trend of preg-robbing with activated carbon compared to other materials. Dash et al. [16] conducted a similar study and reached the same conclusions. Despite the information currently available on preg-robbing by carbonaceous matter, no consensus has been reached on the exact mechanism involved in the preg-robbing of gold.

Carbonaceous matter is not the only material that performs preg-robbing, as silicates are also responsible for reduced recovery rates. Petersen et al. [17] reported that silicates absorb gold from leaching processes, causing losses that can have significant economic impacts. Ferullo et al. [18] analyzed the site of the unbridged oxygen defect of silica, finding that gold showed a minor charge transfer between the metal ions (Cu and Ag) evaluated. Meanwhile, Hong et al.'s [19] work on gold complexes found a solid interaction between gold particles and kaolinite surfaces. Veith et al. [20] determined that gold clumps bind to a single anchor site on the silica surface and that an adequate site can be formed by removing a proton or (OH) group. Wojtaszek et al. [21] explored the mechanism of gold chloride adsorption on mesoporous silica. Their report suggests that adsorption in the case of this surface occurs through hydrogen complexes in an interaction between (Au-Cl) and (Si-OH) bonds on the silica surface. They showed that the (Au-Cl) interaction is stronger than the (Au-OH) interaction.

In other studies, Mohammednejad et al. [22,23] proposed an explanation for this phenomenon, affirming that defects are the leading cause of the formation of interfacial bonds on the surface of silicates. Surface defects can be created by mechanical actions, such as fracturing, crushing, or radiation, but natural mechanochemical activation can also cause this effect in environments where mineral particles are in motion. Wei Sung et al. [9] recognized the existence of discrepancies among results on the preg-robbing of silicates since some do not perform this to any significant degree. Despite all this research, it is still not possible to identify the precise mechanism through which silicates retain gold in cyanidation solutions.

Given the above, the aim of our work was to analyze the degree of preg-robbing by quicklime in gold cyanidation solutions. Quicklime is associated with both species of carbonaceous matter and silicates that, as outlined above, are responsible for preg-robbing and may act synergistically to perform this action. The specific objectives of this study were

to provide an explanation for this phenomenon that will aid in preventing losses in gold cyanidation processes and provide detailed information on an event that has no precedent in the current literature.

## 2. Materials and Methods

Seven quicklime (CaO) samples were used, differentiated by the origin of the raw material and processing conditions. The quicklimes were called vertical kiln (qlvk), rotary kiln (qlrk), and external manufacturer (qlem). The other samples were taken from a rotary kiln, identified, and classified according to their retained 200-mesh Tyler series granulometric fraction. The names assigned were rk + 20%, rk + 5%, rk + 28%, and rk + 35%.

### 2.1. Gold Retention Quantification Test

The tests to quantify the retention of gold in quicklime from the pregnant solution were carried out by placing a sample of quicklime in contact with a certain volume of solution. A bottle was placed in a horizontal position on rotating rollers to provide continuous agitation for 24–72 h. The tests were divided into two large groups, each one given a specific name. Three gold solutions were tested. The pregnant solution obtained by cyanidation—a sample of gold mineral containing sulfides and oxides—was called the natural solution. The second solution was a gold standard solution used to calibrate specific characterization equipment. The third called the synthetic gold solution was manufactured from a 99.9999% purity gold leach in NaCl solution.

The first group of tests was called excess ratio because an excess of quicklime was used with respect to the volume of the solution compared to the proportions used in industrial applications. The experimental conditions were 200 g of lime and 800 mL of total solution. A synthetic gold solution with concentrations of 0.4 ppm and 25 ppm, pH 13, initial cyanide strength of 0.138–0.23, and a burst time of 24 h at 120 RPM was used. Tests T1, T2, and T3 involved a natural solution of 0.4 ppm of gold and a contact time of 72 h, but all other parameters remained the same. Table 1 summarizes the conditions of each test.

**Table 1.** Experimental conditions of the excess ratio tests.

| Quicklime (Qkl) | qlrk (T1) | qlrk (T2) | qlrk (T3) | qlrk (T4) | qlem (T5) | qlvk (T6) | rk + 20% (T7) | rk + 35% (T8) | rk + 5% (T9) | rk + 28% (T10) | qlvk (T11) | qlem (T12) | qlvk (T13) |
|---|---|---|---|---|---|---|---|---|---|---|---|---|---|
| Ratio | | | | | | | Excess | | | | | | |
| pH | 13 | 13 | 13 | 13 | 13 | 13 | 13 | 13 | 13 | 13 | 13 | 13 | 13 |
| Cyanide strength | 0.143 | 0.143 | 0.138 | 0.18 | 0.22 | 0.22 | 0.23 | 0.23 | 0.22 | 0.21 | 0.2 | 0.23 | 0.23 |
| Initial conc. (Au ppm) | 0.459 | 0.455 | 0.442 | 0.451 | 0.451 | 0.451 | 0.45 | 0.45 | 0.45 | 0.45 | 25.36 | 25.36 | 25.32 |

The second set of tests was called the accurate ratio because the ratio of the mass of lime/volume of solution corresponded to that used in industrial cyanidation. Here, 10 g of lime in 500 mL was tested with a synthetic gold solution at 3 concentrations: 5, 15, and 25 ppm of Au. The residence time was 24 h in all tests. The stirring speed was kept constant at 120 rpm, but pH fluctuated in a range of 11–13. The initial cyanide strength (FCN−) varied from 0.18 to 0.25. A standard solution was used for T14, T21, and T28, but all other conditions were identical.

The initial and final concentrations of gold in all tests were measured using ICP-MS (inductively coupled plasma mass spectrometry). Table 2 summarizes the conditions used in these tests.

**Table 2.** Experimental conditions of the accurate ratio tests.

| Qkl | qlrk (T14) | qlrk (T15) | qlem (T16) | rk + 20% (T17) | rk + 5% (T18) | rk + 35% (T19) | rk + 28% (T20) | qlrk (T21) | qlrk (T22) | qlem (T23) | rk + 20% (T24) | rk + 5% (T25) | rk + 35% (T26) | rk + 28% (T27) | qlrk (T28) | qlrk (T29) | qlem (T30) | rk + 20% (T31) | rk + 35% (T32) | rk + 5% (T33) | rk + 28% (T34) |
|---|---|---|---|---|---|---|---|---|---|---|---|---|---|---|---|---|---|---|---|---|---|
| Ratio | Accurate | | | | | | | | | | | | | | | | | | | | |
| pH | 12 | 13 | 13 | 11 | 11 | 11 | 11 | 12 | 13 | 13 | 11 | 11 | 11 | 11 | 12 | 13 | 13 | 11 | 11 | 11 | 11 |
| Cyanide strength | 0.19 | 0.22 | 0.22 | 0.23 | 0.22 | 0.23 | 0.22 | 0.2 | 0.23 | 0.23 | 0.25 | 0.24 | 0.25 | 0.24 | 0.2 | 0.25 | 0.25 | 0.25 | 0.25 | 0.25 | 0.25 |
| Initial conc. (Au ppm) | 5.334 | 5.318 | 5.326 | 5.075 | 5.075 | 5.075 | 5.075 | 15.74 | 15.76 | 15.79 | 15.02 | 15.02 | 15.02 | 15.02 | 25.96 | 25.94 | 25.95 | 25.53 | 25.53 | 25.53 | 25.53 |

## 2.2. Determination of Free Carbon through Acid Digestion

Medium-strength acid digestion was performed to determine the presence of inert carbon in the qlrk and qlvk samples. For digestion, a 300 mL solution of acetic acid ($CH_3COOH$) was used, diluted to 20% in distilled water, and then reacted with 100 g of quicklime. The lime sample was exposed to the acid for 12 h to ensure a complete reaction. The solid residue from the digestion process was then analyzed employing XRD, FTIR, Raman, and carbon analysis by combustion.

## 2.3. X-ray Diffraction (XRD)

The crystalline components of qlrk and qlvk were identified by XRD and Bruker DIFRAC.EVA software. The percentages of the phases obtained were determined semi-quantitatively. The XRD analysis was conducted with a Bruker D2 Phaser diffractometer, with Cu K$\alpha$1 $\lambda° = 1.54060$ Å and radiation and conditions of 30 kV and 10 mA, equipped with an LYNXEYE detector with an amplitude of 5.7° (initial and final position of 2θ from 5–45°).

## 2.4. Carbon and Free Carbon Analyses

An LECO RC612 carbon analyzer was used to determine total carbon concentrations in the qlrk and qlvk residues. Free carbon was measured by stoichiometric calculations and percentage difference.

## 2.5. Fourier-Transform Infrared Spectroscopy—Attenuated Total Reflectance (FTIR-ATR)

FTIR-ATR characterization identified links and functional groups in qlvk. The analysis was conducted using a Perkin Elmer Frontier apparatus with a total attenuated reflection (ATD) diamond accessory. A DTGS detector, Glowbar source, and Csl beam splitter were used. The range measured was 500–1800 cm$^{-1}$ with a resolution of 4 cm$^{-1}$ and 16 scans.

## 2.6. Raman Spectroscopy

This technique was used to find information on the chemical composition, properties, and possible crystallinity or polymorphism of qlvk. Analyses were performed with a Jobin-Yvon LabRam HR 800 dispersive spectrometer coupled to an Olympus BXFM optical microscope with a 532 nm laser line. The size of the laser spot was 2 μm. The diffraction grating was 600 lines/mm, and the laser power was 0.5–5 mW. The objective used was 50×.

## 2.7. Correlation of Silicates and Gold Retention

To correlate the number of silicates with the cyanide tests, the samples were evaluated by X-ray diffraction plus the DIFRAC.EVA software for semiquantitative analysis. The amounts of silicates present without evidence of cyanidation were measured and correlated with the percentage of gold lost in the different solutions. The samples analyzed were qlrk, qlem, qlvk, rk + 20%, rk + 28%, rk + 35%, and rk + 24%.

## 2.8. Morphological/Chemical Analysis of Samples after the Gold Retention Tests

Scanning electron microscopy–energy dispersive spectroscopy (SEM-EDS) analyses were performed using Hitachi SU8230 equipment (Hitachi, Tokyo, Japan) to determine the morphology, structure, and quicklime composition of T14 after the gold retention test. The cold cathode technique was used with the quicklime that was adhered to the sample holder by copper tape. The conditions for taking the images involved applying an energy of 6 kV and 5 µA at a working distance of 2.2 mm.

## 2.9. Time-of-Flight Secondary Ion Mass Spectrometry (TOF-SIMS)

Tests T14, T15, and T16 were selected for TOF-SIMS analyses. Two standard samples were prepared. Both were placed in contact with a 25 ppm gold standard solution for 24 h under roller agitation at 70 rpm. Standard sample 1 contained reagent-grade granulated activated carbon (C), while standard sample 2 had reagent-grade calcium silicate ($2CaO \cdot SiO_2$). A 3:1 mixture of the standard 1 and standard 2 samples prepared as a standard for the SIMS analysis was made with a matrix composition close to that of the samples analyzed. A TOF-SIMS-V spectrometer (IonTOF Co., Münster, Germany) was used in all experiments. Analyses were performed by pulsing $Bi^{3+}$ bismuth ions with an energy of 30 keV. The ion current was approximately 0.2–0.3 pA, and the ion beam was spread out in a scan with dimensions between $70 \times 70$ and $200 \times 200$ microns. The surface positive charge from ionic irradiation was compensated by an electron beam with an energy of 20 eV and a current of ~12 µA.

The composition of the sample surface was analyzed in the mass spectrum measurement mode. The distribution of the different elements on the surface was analyzed in the ion imaging mode (chemical map of the surface). Secondary negative ions were used as analytical signals. The residual pressure in the analysis chamber during the measurements did not exceed 9–10 Torr. For the SIMS analysis, samples were prepared in pellet form using ultra-pure indium as a matrix. The powders were poured into a thin, even layer and indium-pressed (by hand) using a metal cylinder, and then, the surface was covered with a sheet of ultra-pure molybdenum.

## 3. Results and Discussion

### 3.1. Gold Loss in the Excess Ratio Tests

Figure 1 shows the gold losses in the excess ratio tests. The average percentage loss was 6.784%, while the total average loss in ppm was 0.446. The T1, T2, and T3 samples that used the natural solution did not show results significantly different from the other tests. Likewise, the samples with different granulometry did not show significantly different results from those with normal granulometry. This phenomenon could explain why, although there are different contact surfaces, preg-robbing would occur anyway. Increasing the proportion of lime, which interacts with the gold to precipitate it (presumably due to activated embedded carbon/silicates), may also have increased gold loss in the solution. The higher proportion of quicklime compared to the levels used in industrial cyanidation was responsible for the rate of gold retention.

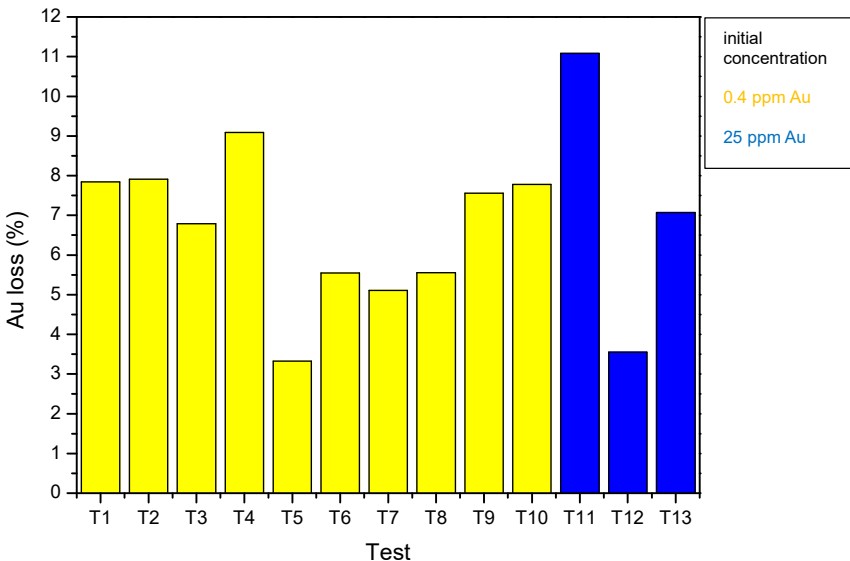

**Figure 1.** Au loss (%) in the excess ratio tests.

### 3.2. Gold Loss in the Accurate Ratio Tests

Figure 2 presents the results of the accurate ratio tests. As described above, the tests with different granulometry did not show different results. There was an average gold retention of 0.981% in these samples, while total loss averaged 0.149 ppm. The average percentage considering the two test groups together was 3.135%, while the ppm loss was 0.255. There was a percentage difference of 5.803% between the excess ratio and accurate ratio test conditions and in terms of total loss, which was 0.297 ppm. The ratio of lime to the volume of solution used in these tests is comparable to that used in cyanidation processes. The fact that considerable losses continue to occur is significant because, in an important industry like gold extraction, process efficiency must be very high to prevent substantial, long-term economic losses. These tests were designed to detect preg-robbing, a phenomenon that may be attributable entirely to quicklime, with no possibility of participation by other species.

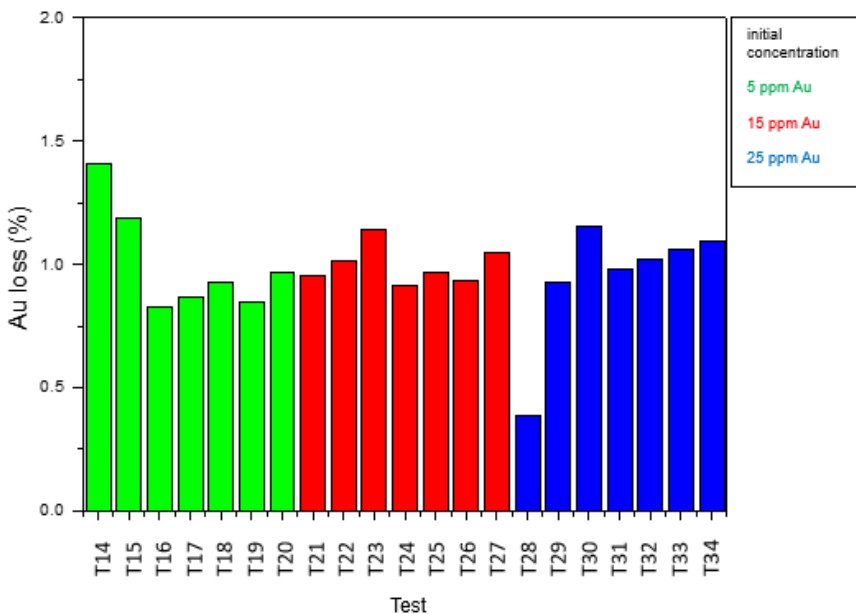

**Figure 2.** Au loss (%) in the accurate ratio tests.

### 3.3. Ratio of Gold Retained to the Mass of Quicklime Used (mg Au/kg-ql)

Having obtained the gold loss data from each test, the mg Au/kg-ql ratio (milligrams of gold/kilogram of quicklime) was measured and quantified using the following formula:

$$\frac{Mg_{Au}}{kg_{ql}} = \frac{(f_cAu - i_cAu) \times (Vol.sln)}{m_{cal}}$$

(2)

where

$i_cAu$ = initial gold concentration in ppm;
$f_cAu$ = final gold concentration in ppm;
Vol. sln = volume of solution in L;
mcal = mass of quicklime in kg.

### 3.4. The mg Au/kg-ql Ratio in the Accurate Ratio Tests

Figure 3 shows the losses of milligrams of gold per kilogram of quicklime from the excess ratio samples with 0.4 ppm Au initial concentration and Figure 4 with 25 ppm Au initial concentration obtained with Formula (2). The average loss was 1.784 mg Au/kg ql. There were no significant differences between the samples with the natural solution and those with distinct granulometry. As can be seen in Figures 3 and 4, only the tests with an initial concentration of 25 ppm had much more significant losses than those with 0.4 ppm in all cases. This test implies that in the case of mg Au/kg, retention is based on the initial gold concentration since more gold is available for preg-robbing.

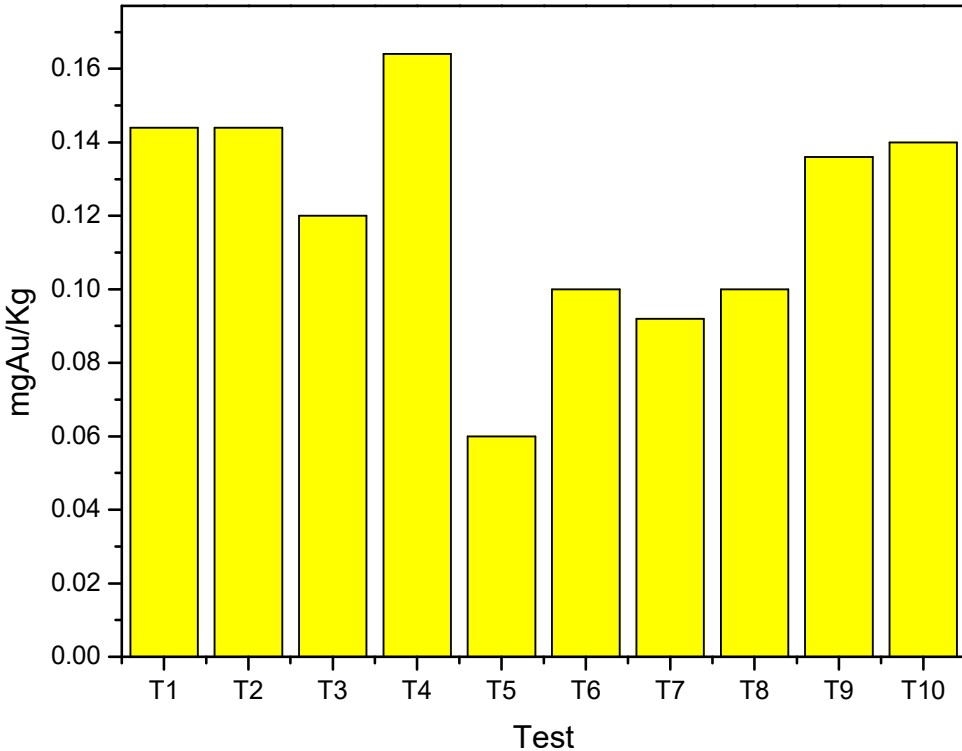

**Figure 3.** mg Au/kg-ql loss in the excess ratio tests (0.4 ppm Au initial concentration).

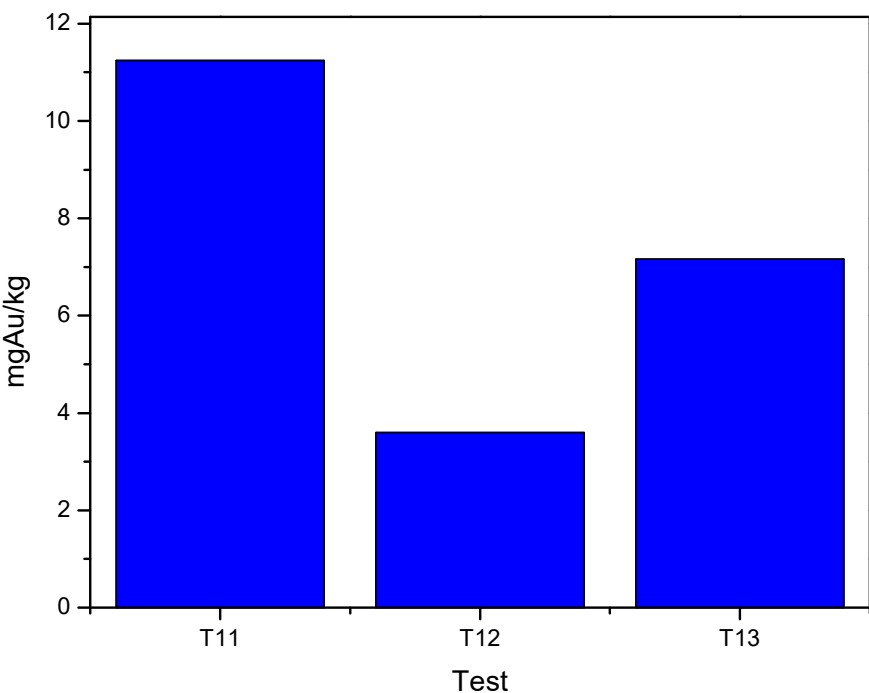

**Figure 4.** mg Au/kg-ql loss in the excess ratio tests (25 ppm Au initial concentration).

*3.5. The mg Au/kg-ql Ratio in the Excess Ratio Tests*

Figure 5 displays the mg Au/kg-ql from the accurate ratio tests. The average loss was 7.485 mg Au/kg-ql. Once again, no differences among the tests with distinct granulometries were observed. There was a 5.701 mg Au/kg-ql difference between the two groups of tests, with the accurate ratio showing a greater mg Au/kg-ql loss than the excess ratio. A tendency in the mg Au/kg-ql ratio seen in both groups of assays dictates that the tests with the highest concentration of gold registered the most significant losses, measured in milligrams. Thus, the higher the concentration of gold during cyanidation, the greater the degree of preg-robbing.

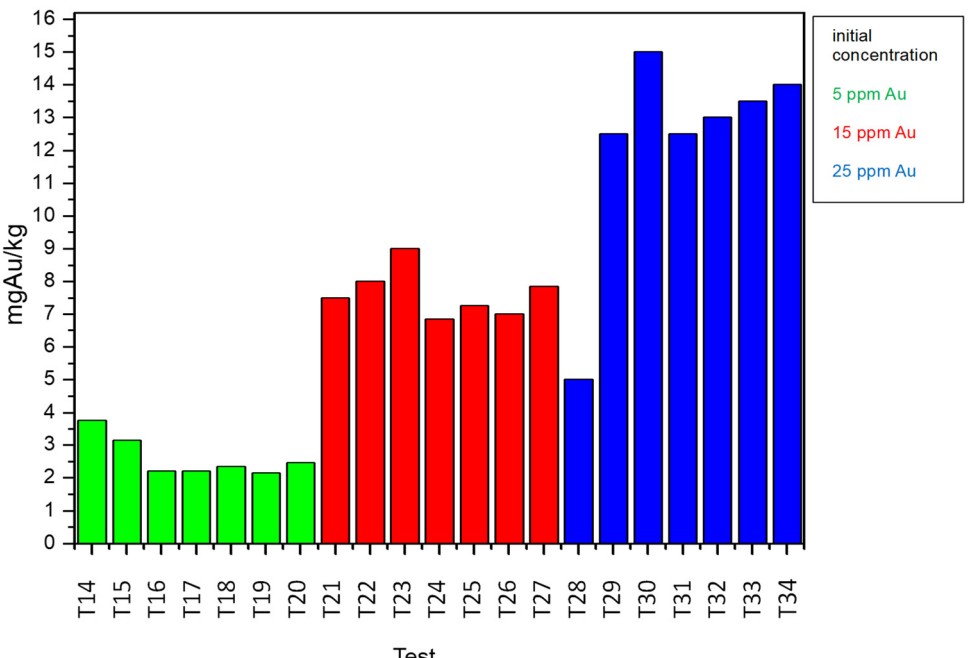

**Figure 5.** mg Au/kg-ql loss in the accurate ratio tests.

Another trend was that the amount of gold retained in mg Au/kg-ql proved to be inversely proportional to the ratio of quicklime in the volume of solution. The results indicate that more gold was retained in the accurate ratio tests than in the excess ratio tests. Regarding the percentage of gold loss, this value increased with higher concentrations of quicklime.

### 3.6. Determination of Free Carbon through Acid Digestion

The purposes of performing the acid digestion of quicklime (CaO) samples are to (i) eliminate the carbon associated with carbonate and (ii) reduce the sample weight in order to concentrate carbonaceous species of organic origin and then determine them. Acetic acid is a medium-to-low-strength digester, so the potential for attack on carbonaceous components is lower. This process occurs according to the equation in which the reaction of acetic acid ($CH_3COOH$) and calcium carbonate ($CaCO_3$) produces calcium acetate ($(CH_3COO)_2Ca$), water ($H_2O$), and carbon dioxide ($CO_2$).

$$2CH_3COOH + CaCO_3 = (CH_3COO)_2Ca + H_2O \uparrow + CO_2 \uparrow \qquad (3)$$

### 3.7. XDR

Figure 6 shows the X-ray diffraction analysis of qlvk and qlrk. The main phase detected consisted of hydrated calcium acetate [$(CH_3COO)_2Ca \cdot H_2O$], which diffracted in most of the peaks, especially at 7.30°, 10.72°, 11.88°, 14.51°, 15.95°, 23.45°, 25.95°, and 27.62° 2θ, according to the reference pattern 00-019-0199 [24]. Other phases identified were ikaite ($CaCO_3 \cdot 6H_2O$) at 17.19° 2θ [25] and brianroulstonite [$Ca_3[B_5O_6(OH)_6](OH)Cl_2 \cdot 8H_2O$], with a main peak at 10.92° 2θ [26]. Calcite ($CaCO_3$) diffracted at an angle of 29.48° 2θ [27], while ettringite [$Ca_6Al_2(SO_4)_3(OH)_{12} \cdot 26H_2O$] presented a main peak at 9.76° 2θ [27]. Other peaks detected were bayerite [$Al(OH)_3$] at 27.70 2θ [28] and quartz ($SiO_2$) at 20.22° 2θ [29]. Some peaks were not verified completely, perhaps due to "noise" detected by the equipment.

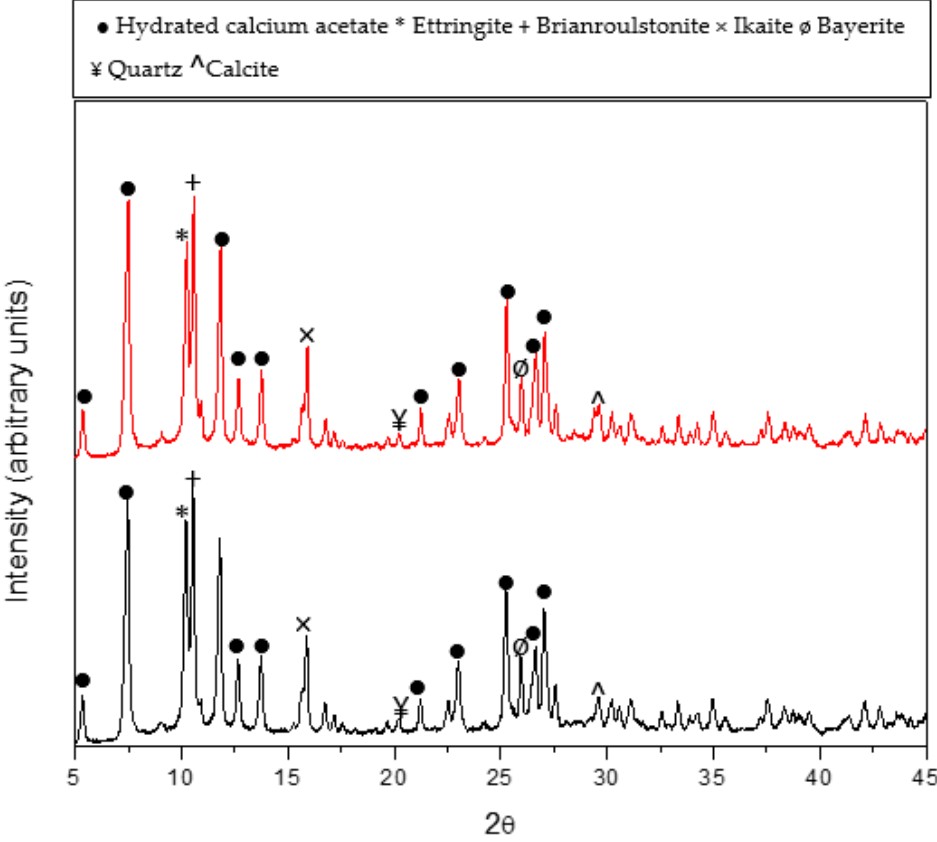

**Figure 6.** XRD analysis of the qlvk and qlrk samples.

### 3.8. Carbon and Free Carbon Analysis

The semiquantitative analysis made it possible to determine the percentages of the mineral phases in qlrk and qlvk, as well as total carbon, obtained by elemental carbon analysis. The results are shown in the table. Three compounds—hydrated calcium acetate [(CH$_3$COO)$_2$Ca·H$_2$O], ikaite (CaCO$_3$.6H$_2$O), and calcite (CaCO$_3$)—had a carbon (C) composition. The percentage of carbon in each one was determined by stoichiometric calculations, and then, the three percentages were totaled to determine the percentage that was chemically bound (%C). These calculations were made as follows:

Hydrated calcium acetate [(CH$_3$COO)$_2$Ca·H$_2$O]

$$\%C = [(\%(CH_3COO)_2Ca·H_2O)]\,[M.\text{ weight}(4C)]\,[M.\text{ weight }(CH_3COO)_2Ca·H_2O] \tag{4}$$

$$\%C = [\%(CH_3COO)_2Ca·H_2O\,[48]\,[176.17] \tag{5}$$

Ikaite CaCO$_3$·6H$_2$O

$$\%C = [\%\,CaCO_3·6H_2O]\,[M.\text{ weight }(C)]\,[M.\text{ weight }CaCO_3·6H_2O] \tag{6}$$

$$\%C = [\%CaCO_3·6H_2O]\,[12]\,[208] \tag{7}$$

Calcite CaCO$_3$

$$\%C = [\%\,CaCO_3]\,[M.\text{ weight }(C)]\,[M.\text{ weight }CaCO_3] \tag{8}$$

$$\%C = [CaCO_3]\,[12]\,[100] \tag{9}$$

These calculations were used to measure the percentage of chemically bound C in qlvk and qlrk according to Table 3. These values were then subtracted from the total %C to calculate free %C. For qlvk, free %C was 2.853, while for qlrk it was 3.948. Calculating the % of free C is essential because it could be responsible for gold retention, since it does not bond to other species and, hence, can form adsorption bonds in cyanidation solutions.

**Table 3.** Phases and free carbon in the qlvk and qlrk samples.

| Phase | qlvk (%) | qlrk (%) |
|---|---|---|
| Hydrated calcium acetate | 73.3 | 73.4 |
| Ikaite | 10 | 10 |
| Brianroulstonite | 7.2 | 6.1 |
| Calcite | 6.7 | <1 |
| Ettringite | 2.7 | <1 |
| Bayerite | <1 | 4.9 |
| Quartz | <1 | 3.5 |
| Total %C | 26.6 | 26.8 |
| Chemically bound %C | 23.746 | 22.851 |
| Free %C | 2.853 | 3.948 |

During calcination, the carbonaceous matter in the limestone was exposed to temperature and gaseous atmosphere conditions that favored physical activity. This results in the potential formation of activated carbon, which has a high gold sorption capacity. However, since this form of carbon has no defined formula or characteristics, it can be obtained either from raw materials or through physical and chemical activation processes [30]. Considering that limestone treated by high-temperature thermolysis contains carbonaceous matter, the results could have similarities to the physical activation process used to produce activated carbon. Physical activation involves the reaction between samples and gas

(CO$_2$ or air), steam, or a gas mixture at temperatures >700 °C [31,32]. These similarities between treatments may explain the possible transformation of carbonaceous matter into activated carbon. During the physical activation of activated carbon, the disordered carbon in the spaces of the most reactive regions of the char is exposed to oxygen and undergoes endothermic reactions. Water vapor (H$_2$O) reacts with the carbon structure to produce carbon monoxide (CO), which is accompanied by the transformation of water into gas, as shown in Equations (10)–(12) [33] (see below).

$$C + H_2O = CO + H_2 \; \Delta H_{1073K} = 136 \text{ kJ (800 °C)} \tag{10}$$

$$C + CO_2 = 2CO \; \Delta H_{1073K} = 170 \text{ kJ (800 °C)} \tag{11}$$

$$CO + H_2O = CO_2 + H_2 \; \Delta H_{1073K} = -34 \text{ kJ (800 °C)} \tag{12}$$

*3.9. FTIR-ATR Analysis*

Figure 7 shows the FTIR analysis of qlvk. The 616 cm$^{-1}$, 642 cm$^{-1}$, 659 cm$^{-1}$, 671 cm$^{-1}$, 946 cm$^{-1}$, 1030 cm$^{-1}$, 1051 cm$^{-1}$, 1413 cm$^{-1}$, 1446 cm$^{-1}$, 1467 cm$^{-1}$, 1528 cm$^{-1}$, and 1604 cm$^{-1}$ bands coincided with hydrated calcium acetate [24]. At 659 cm$^{-1}$ and 671 cm$^{-1}$, the symmetric twisting and rocking vibration of the O–C–O fragment took place, while the stretching vibration of C–C bonding was observed at 946 cm$^{-1}$. At 1030 cm$^{-1}$, the in-plane bending vibration of methyl (CH$_3$) occurred, and at 1051 cm$^{-1}$, the out-plane stretching vibration of methyl (CH$_3$) and 1413 cm$^{-1}$ methyl (CH$_3$) proceeded antisymmetrically. Finally, 1446 cm$^{-1}$ was the point where the symmetric vibration of C–O occurred. One result that stood out was the detection of carbon with distinct types of bonds and elements, as this suggested the presence of activated carbon, a species with preg-robbing potential.

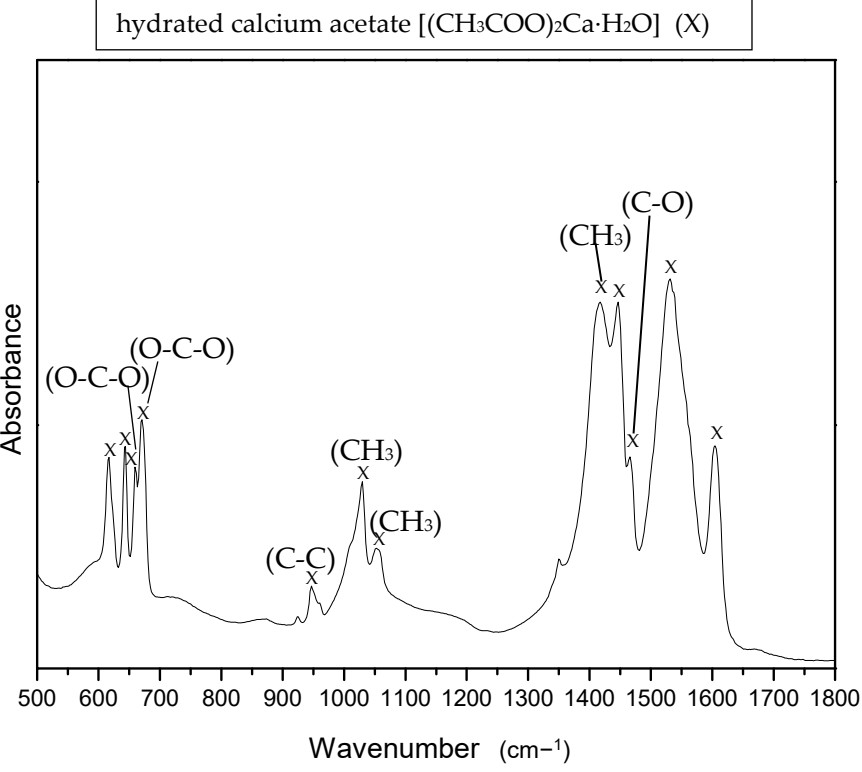

**Figure 7.** FTIR analysis of qlvk.

*3.10. Raman Spectroscopy*

Figure 8 shows the Raman spectra of qlvk. The analysis verified the typical spectrum of hydrated calcium acetate [(CH$_3$COO)$_2$Ca·H$_2$O] in the Raman shift at 671 cm$^{-1}$, 965 cm$^{-1}$,

1365 cm$^{-1}$, 1438 cm$^{-1}$, 1473 cm$^{-1}$, and 2932 cm$^{-1}$ for all the quicklime samples [24]. These findings coincide with those from the FTIR test, where calcium acetate was also found. In addition, the functional group of O–C–O was found symmetrically at 671 cm$^{-1}$. Similarly, methyl (CH$_3$) was found at 2932 cm$^{-1}$. Within the functional groups detected, the element (C) was present in combination with others. Remarkably, functional group distinctions between Raman and FTIR-ATR spectroscopy were discovered. This is indicative of the presence of carbonaceous matter that may exist in the form of activated carbon after a process of limestone thermolysis. This characterization was added to the previous ones, where species with preg-robbing potential existed, but (Au) was not detected.

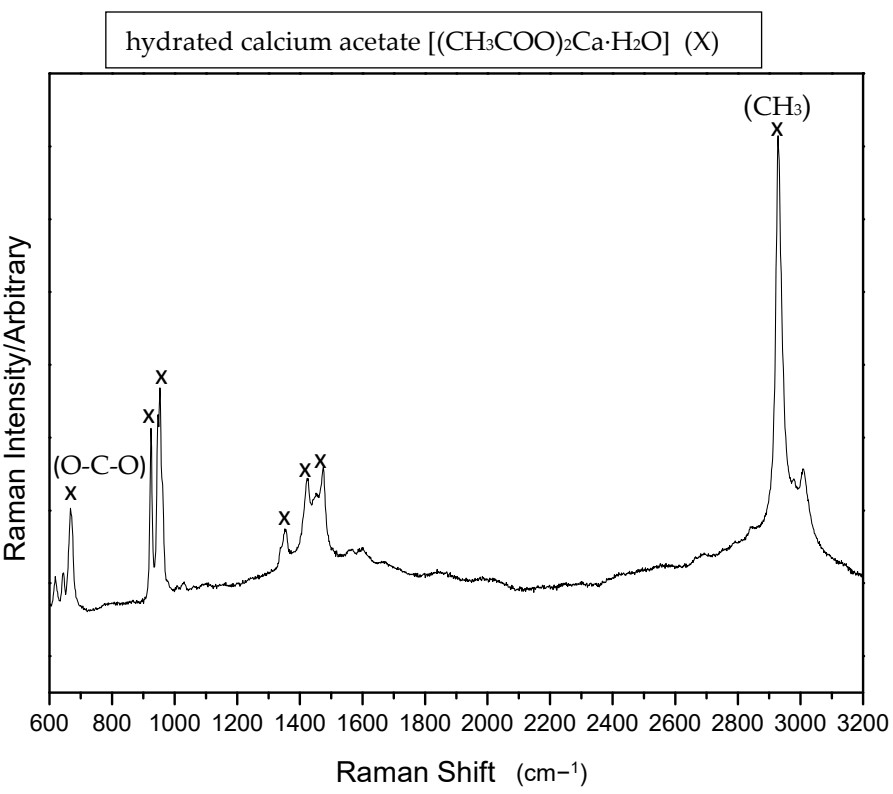

**Figure 8.** Raman spectra of qlvk.

### 3.11. Correlation of Gold and Silicate Retention

Table 4 shows the results of the X-ray diffraction analysis conducted with DIFRA.EVA software to extract the phases of the qlrk, qlvk, qlem, rk + 20%, rk + 28%, rk + 35%, and rk + 24% samples. Compounds such as calcium oxide (CaO) were primarily found in all samples, together with traces of portlandite (Ca(OH)$_2$), periclase (MgO), quartz (SiO$_2$), and calcite (CaCO$_3$). Species such as C$_3$S (tricalcium silicate 3CaO-SiO$_2$), C$_2$S (dicalcium silicate 2CaO-SiO$_2$), C$_3$A (tricalcium aluminate 3CaO-Al$_2$O$_3$), and C$_4$A (tricalcium aluminum ferrite 4CaO-Al$_2$O$_3$-FeO$_2$) were also detected. The presence of two silicate compounds C$_3$S (tricalcium silicate 3CaO-SiO$_2$) and C$_2$S (dicalcium silicate 2CaO-SiO$_2$) is important as they could perform preg-robbing or influence this process in some way. C$_2$S (dicalcium silicate 2CaO-SiO$_2$) was correlated with the loss percentages in the gold cyanidation samples. The tests chosen for this correlation analysis were from the excess ratio assays (Table 1). An average of the test results was used for qlrk, qlvk, and qlem.

**Table 4.** Minerals present in the samples in weight %.

| Phase | qlrk | qlem | qlvk | rk + 20% | rk + 28% | rk + 35% | rk + 24% |
|---|---|---|---|---|---|---|---|
| $C_3S$ (tricalcium silicate $3CaO\text{-}SiO_2$) | 0.67% | 1.07% | 1.18% | 0.39% | 0.71% | 0.57% | 0.73% |
| $C_2S$ (dicalcium silicate $2CaO\text{-}SiO_2$) | 15.57% | 5.82% | 8.5% | 10.24% | 12.78% | 12.97% | 14.13% |
| $C_3A$ (tricalcium aluminate $3CaO\text{-}Al_2O_3$) | 1.77% | 0.8% | 0.65% | 0.6% | 1.11% | 0.76% | 1.14% |
| $C_4AF$ (tetracalcium aluminoferrite $4CaO\text{-}Al_2O_3\text{-}FeO_2$) | 1.6% | 1.67% | 1.24% | 2.05% | 1.45% | 1.98% | 1.63% |
| Calcium oxide (CaO) | 78.06% | 82.14% | 80.62% | 81.17% | 81.14% | 80.91% | 77.64% |
| Portlandite ($Ca(OH)_2$) | 0.17% | 0.41% | 0.07% | 0.13% | 0.23% | 0.08% | 1.08% |
| Periclase (MgO) | 1.64% | 1.51% | 1.45% | 1.47% | 1.8% | 1.73% | 1.79% |
| Quartz ($SiO_2$) | 0% | 0.07% | 0.4% | 0.14% | 0.14% | 0.12% | 0.25% |
| Calcite ($CaCO_3$) | 0.52% | 6.51% | 5.89% | 3.81% | 0.64% | 0.88% | 1.61% |

Figure 9 shows the following correlation: as the amount of $C_2S$ (dicalcium silicate $2CaO\text{-}SiO_2$) in the qlrk and qlvk samples increased, the loss of gold also increased, while a decrease in $C_2S$ resulted in reduced loss, as occurs in the case of qlem. With respect to rk + 20%, rk + 28%, rk + 35%, and rk + 24%, there were slight variations between the two values analyzed.

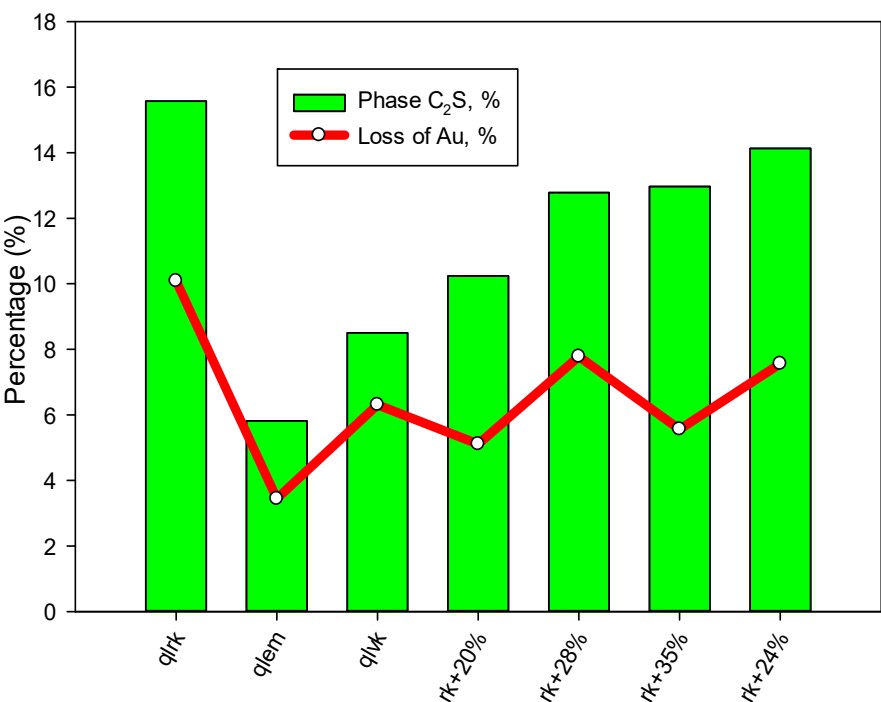

**Figure 9.** Correlation between the $C_2S$ phase of excess ratio and gold losses (%).

The processes involved in transforming limestone to quicklime and manufacturing clinker for cement are quite similar, so this could explain the formation of calcium silicate in quicklime. Initially, all the contained moisture evaporated at 100–200 °C, while the formation of the oxides $SiO_2$, $Al_2O_3$, and $Fe_2O_3$ began between 400 and 750 °C. At temperatures

above 800 °C, calcium oxide is produced from calcium carbonate in a process known as decarbonization or calcination:

$$(CaCO_3 \rightarrow CaO + CO_2) \tag{13}$$

Up to ~1200 °C and in the solid phase, the formation of compounds with a low lime content begins, produced by clay and limestone oxides:

$$(2CaO \cdot SiO_2), (3Ca \cdot Al_2O_3) \text{ and } (4CaO \cdot Al_2O_3 \cdot Fe_2O_3) \tag{14}$$

Between 1260 and 1450 °C, clinkerization occurs, and the so-called molten phase ($3CaO \cdot SiO_2$) forms. The clinker is then cooled in a kiln until it reaches at least 1200 °C [34]. Although the limestone thermolysis processes and the manufacture of clinker for cement are not exactly the same, they are quite similar. This similarity indicates that the formation of silicates in quicklime is possible, both chemically and thermodynamically.

### 3.12. Morphological/Chemical Analysis after Cyanidation

Scanning electron microscopy was used to detect evidence of preg-robbing. The morphological analysis corresponds to T14 from different zones such as Figure 10 zone (a), Figure 11 zone (b) and Figure 12 zone (c). The results show that circular structures tended to form clusters with spaces between them in zones (a) and (b), while in zone (c), circular formations appeared with no apparent space between them. The results of the EDS analysis shown in Table 5 present the percentages of each element in the sample.

**Table 5.** EDS analysis of T14 in zones (a), (b), and (c).

| Element | Zone (a) | Zone (b) | Zone (c) |
|---------|----------|----------|----------|
| Carbon | 6.8% | 8.3% | 6.8% |
| Chlorine | 0.2% | 0.3% | 0.5% |
| Aluminum | 0.3% | 0.6% | 0.8% |
| Silicon | 1% | 1.5% | 1.7% |
| Magnesium | 0.6% | 1% | 1.1% |
| Calcium | 31.4% | 27.7% | 28% |
| Oxygen | 59.5% | 60.4% | 60.7% |

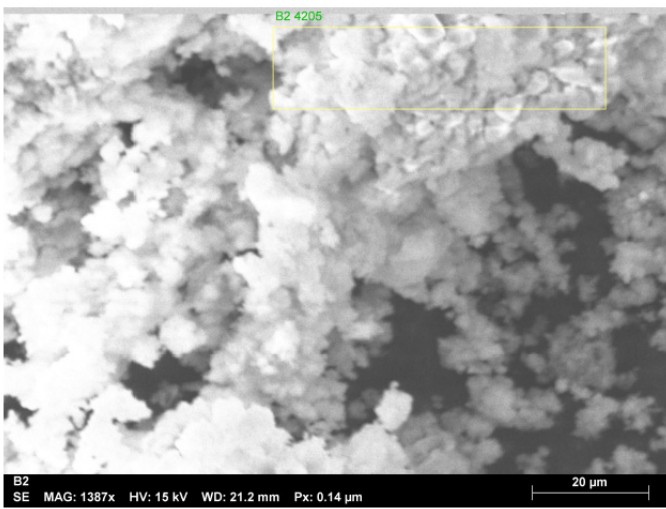

**Figure 10.** Scanning electron microscopy of T14 in zone (a).

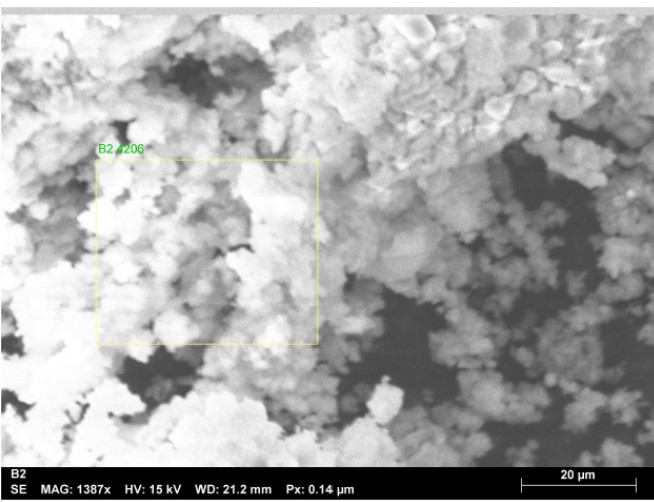

**Figure 11.** Scanning electron microscopy of T14 in zone (b).

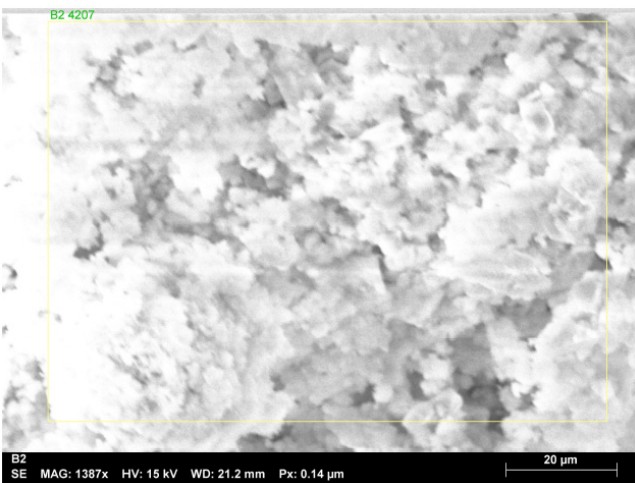

**Figure 12.** Scanning electron microscopy of T14 in zone (c).

These results show that elements, such as oxygen and calcium, were found primarily in the three zones, as expected. Carbon (C) in zones (a) (6.8%), (b) (8.3%), and (c) (6.8%) was related to the carbonaceous matter that could be transformed into activated carbon. In addition, silicon (Si) in zones (a) (1%), (b) (1.5%), and (c) (1.7%) could combine with oxygen to form silicates, two species that share the potential to cause preg-robbing. This technique, however, cannot detect trace elements such as gold, so another method was required to reveal more information, in this case, TOF-SIMS.

### 3.13. TOF-SIMS Analysis

Figure 13 shows a fragment of the negative ion mass spectrum (from 1–350 a.m.u.) of the T16 sample. The mass spectra for the other two samples are almost identical. Slight differences are visible in the intensities of the peaks that correspond to distinct elements and clusters (fragments of molecules). In addition to the main components (H, C, O, Si, and Ca), the figure shows some clusters with maximum intensity. Elements indicative of contamination, such as F, Cl, and Mo, are also observed among the peaks. Molybdenum was clearly detected because molybdenum foil was used to embed the samples in indium. The complex composition of the targets made it impossible to identify all the peaks in the mass spectra, but this task was not included in the study protocol. For obvious reasons, special attention was paid to identifying the gold peak and those that corresponded to the compounds of gold atoms in the elemental form (Au) or compound Au(CN), Au(CN)$_2$, as

well as elements that had the potential to perform preg-robbing, that is, carbon calcium oxide and silicates.

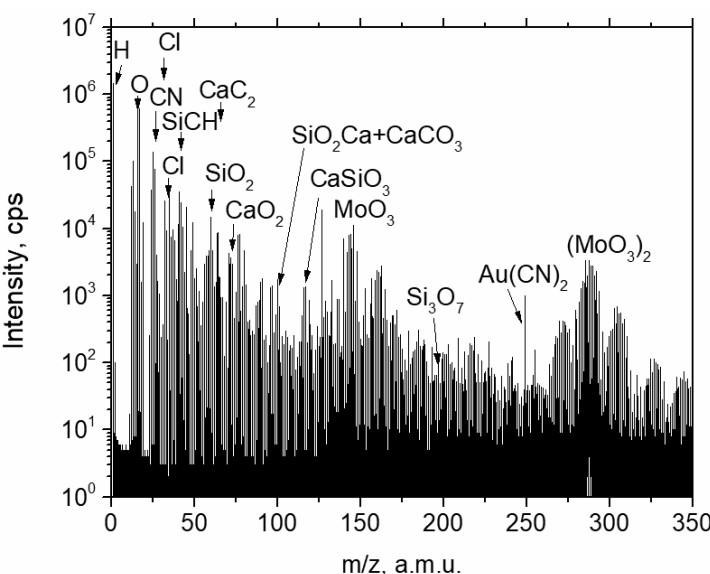

**Figure 13.** Negative ion TOF-SIMS spectrum of sample T16.

Figure 14 shows fragments of the mass spectra for the three samples, T14, T15, and T16, compared to the standard for masses near mass 197 a.m.u. (Au) (a), mass 223 a.m.u. (AuCN) (b), and mass 249 a.m.u. (Au(CN)$_2$) (c). Fragments for mass 253 a.m.u. (AuCaO) and mass 241 a.m.u. (AuSiO) are not given since no corresponding peaks were detected. Other gold-containing clusters were also not detected.

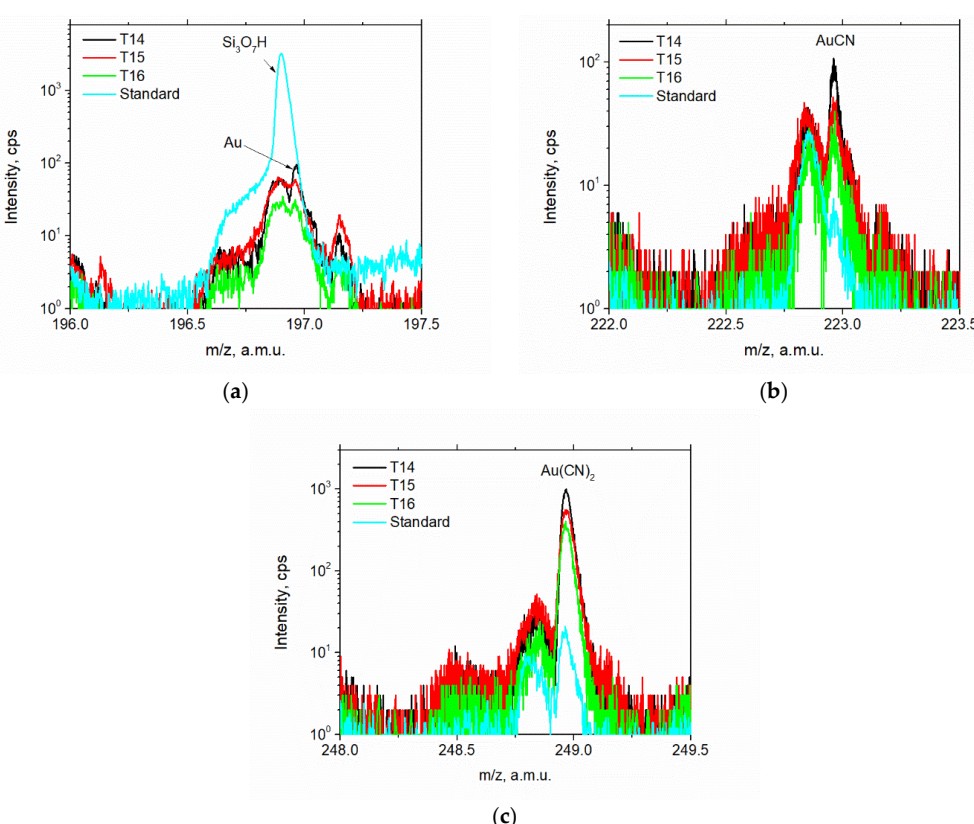

**Figure 14.** Fragments of negative TOF-SIMS mass spectra for samples T14, T15, and T16, and the standard of the masses 197 a.m.u. (**a**), 223 a.m.u. (**b**), and 249 a.m.u. (**c**).

These findings lead to the conclusion that stable $Au(CN)_2-$ complexes were detected in Figure 14c in all three samples (T14, T15, T16). No clusters containing gold atoms bonded to SiO or CaO were observed. The distributions of $Au-$, $AuCN-$, and $Au(CN)_2-$ were similar, suggesting the formation of a stable gold cyanide compound ($Au(CN)_2$).

*3.14. Ionic Images*

Figure 15 shows the lateral distributions of certain secondary cluster ions detected ($CN-$, $CaO_2-$, $SiO_2-$, and $Au(CN)_2-$) on the surface of all three samples. A correlation in the distribution of $CN-$ and $Au(CN)_2-$ over the sample surface is visible. (The $AuCN-$ signal shows the same distribution as $Au(CN)_2-$.) At the same time, the distribution of the secondary cluster ions ($SiO_2-$ and $CaO_2-$) shows the "opposite" distribution; that is, in the surface areas with the maximum $Au(CN)_2-$ signal, the signals for the cluster ions $SiO_2-$ and $CaO_2-$ are minimal and inverted. This means that the distribution of gold inclusions correlates with cyanide but not with silicate and calcium oxide. These findings highlight the fact that the TOF-SIMS and mass analyses allowed us to detect Au and other, related, stable complexes, as well as species with preg-robbing potential, a significant advance that was not possible in previous characterizations.

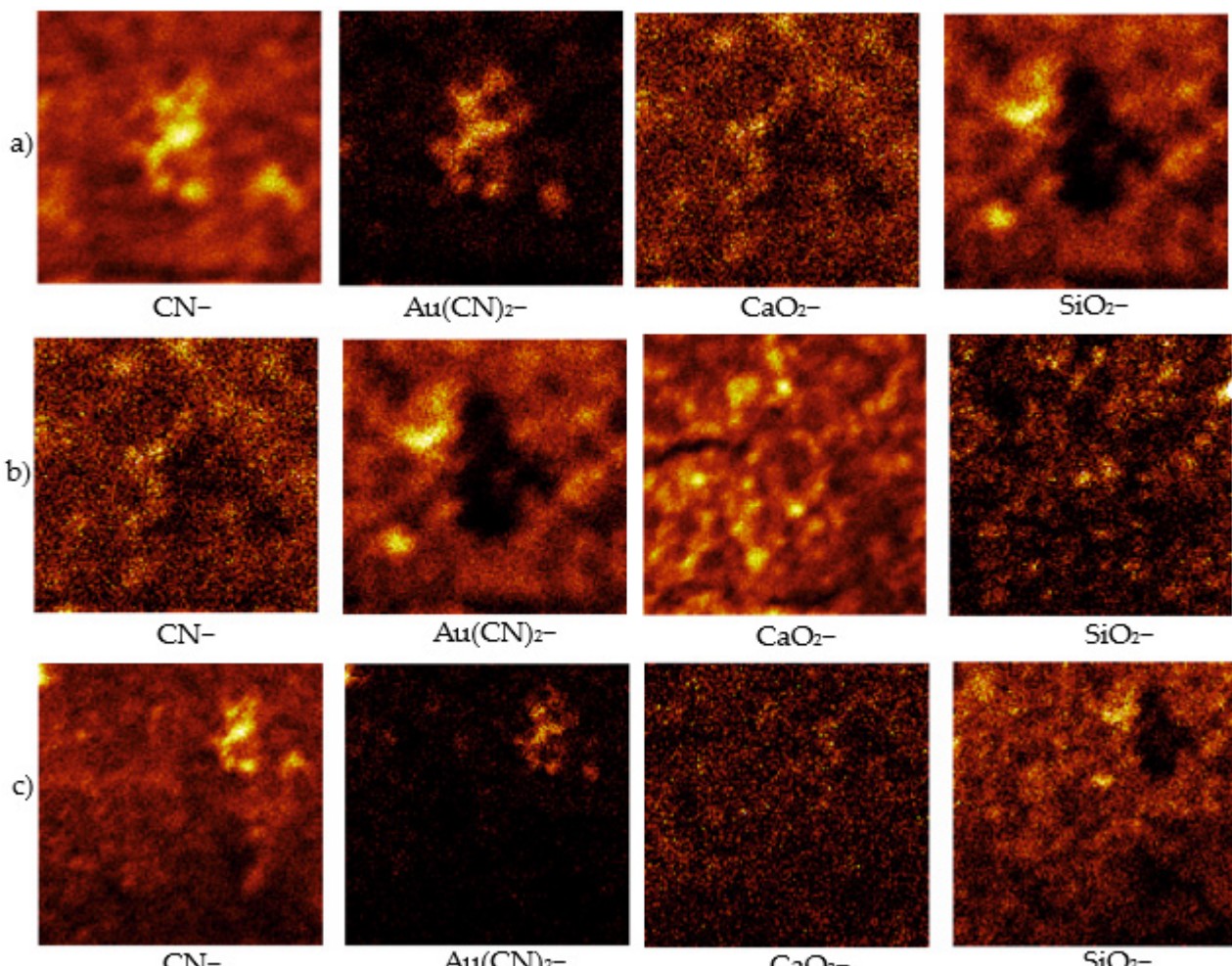

**Figure 15.** Secondary ion images of some cluster ions for samples (**a**) T14, (**b**) T15, and (**c**) T16. Image size is $250 \times 250$ microns ($256 \times 256$ pixels).

*3.15. Chemisorption*

Chemisorption is a type of adsorption that involves a chemical reaction between the surface and the adsorbate in which new chemical bonds are generated on the surface of the

adsorbent. The intense interaction between the adsorbate and the substrate surface creates new types of electronic bonds. In our study, gold mixed with sodium cyanide produced the following Elsener reaction:

$$4Au + 8\,NaCN + O_2 + 2\,H_2O \rightarrow 4\,NaAu(CN)_2 + 4\,NaOH \tag{15}$$

In the leaching process, sodium cyanide is added to the gold ore. Since gold is soluble after this process, it can move freely through the solution, while all other minerals can neither pass through nor form ligands with the gold.

The gold-pregnant solution formed was treated with zinc and activated carbon to extract the gold. Cementation (Merrill–Crowe process) is a process that involves using a zinc electrode with an activated carbon paste immersed in a solution containing gold cyanide. This refining process may share some similarities with TOF-SIMS analysis, where the element (Au) is observed. A similar reaction could be characterized as follows:

$$e- + [Au(CN)_2]- \rightarrow Au + 2CN- \tag{16}$$

At the cathode, gold is reduced by gaining electrons or decreasing the oxidation number, as shown in Equation (16). This reaction could be directly related to the electrochemical reactions that occur in the cyanidation of gold at an industrial level, thus demonstrating that bottle cyanidation tests may offer an accurate notion of the processes and reactions that would occur in cyanidation at that level. Finally, this discussion of chemisorption shows that these two mechanisms might coincide.

*3.16. Physisorption*

Physisorption occurs due to weak, generally non-specific, forces, such as van der Waals and London forces. It tends to reach equilibrium quickly, has very low activation energies, and is reversible. Some authors distinguish two types of physical adsorption: van der Waals adsorption, which occurs when the substance remains mainly on the surface of the adsorbent (e.g., in the case of impurities in activated carbon) and cases where the substance seems to penetrate to an appreciable depth.

In the case of adsorption by layers—specifically graphitic layers in the material—adsorption occurs through physical phenomena, mainly due to Van der Waals's attractive forces. Since the molecules do not share or transfer electrons, both $(AuCN)-$ and quicklime maintain their individuality. Physical adsorption is not specific and generally progresses toward graphitic monolayers or the formation of multilayers.

In the 1920s, Gross and Scott conducted one of the first detailed investigations of gold and silver cyanide adsorption based on activated carbon. They suggested the mechanism specified below for the adsorption of gold on activated pinewood carbon containing $Ca(OH)_2$ but discovered that 50% of the ash from this coal was CaO, so the following reaction took place [35]:

$$2K[Au(CN)_2] + Ca(OH)_2 + CO_2 \leftrightarrow Ca[Au(CN)_2]_2 + 2KHCO_3 \tag{17}$$

It is important to describe Gross and Scott's reaction because the activated carbon they analyzed contained essential amounts of calcium oxide (CaO), which produces a response that could be described in a way quite similar to what occurred in the retention of gold by quicklime. This reaction appears to be reversible. Although potassium (K) was present in the system, sodium cyanide (NaCN) was used in the gold retention process with quicklime. Therefore, the reaction could be adapted as follows:

$$2Na[Au(CN)_2] + Ca(OH)_2 + CO_2 \leftrightarrow Ca[Au(CN)_2]_2 + 2NaHCO_3 \tag{18}$$

## 4. Conclusions

The XRD, Raman, and FTIR-ATR characterization tests all showed hydrated calcium acetate $(CH_3COO)_2Ca \cdot H_2O$ and other common compounds in quicklime. The Raman and FTIR-ATR procedures revealed, especially, carbon functional groups with other elements, while X-ray diffraction confirmed the presence of silicates like $C_3S$ (tricalcium silicate 3CaO-$SiO_2$) and $C_2S$ (dicalcium silicate 2CaO-$SiO_2$). The presence of silicates and carbonaceous matter in the chemical composition of quicklime is highlighted here because both of these species can cause preg-robbing.

Bottle cyanidation tests dictate that preg-robbing is attributable to the action of quicklime due to the isolation of the system. The amount of quicklime present influences the percentage of gold retention because the excess ratio had a higher value than the accurate ratio. At the same time, the mg Au/kg quicklime ratio was inversely proportional, as the accurate ratio test retained more mg Au/kg quicklime than the excess ratio assay. These retention tests also indicate that the amount of gold present was high, which means that the losses could be potentially substantial.

The participation of carbonaceous matter in gold cyanidation is feasible due to a possible conversion into activated carbon through thermolysis that transforms limestone into quicklime in a process that may have similarities to the physical activation of activated carbon. The transformation of carbonaceous matter into limestone could produce activated carbon in the quicklime. Based on the evidence that activated carbon interacts with gold, this would explain a certain proportion of the losses that occurred during the cyanidation process.

An increase in the number of silicates present in quicklime, especially $C_2S$ (dicalcium silicate 2CaO-$SiO_2$), can be correlated with an increase in the percentage of gold loss. The formation of various types of silicates in quicklime is possible due to similarities with the manufacture of cement clinker. The negative ion mass spectrum revealed the main components (H, C, O, Si, and Ca). The results show fragments of mass spectra for the samples analyzed compared to the standard for masses close to (Au), $(AuCN)-$, and $(Au(CN)_2)$, suggesting the formation of a stable gold cyanide compound $(Au(CN)_2)$. The lateral distributions detected secondary cluster ions, such as $CN-$, $CaO_2-$, $SiO_2-$, and $Au(CN)_2-$, on the surface of the samples. A correlation in $CN-$ and $Au(CN)_2-$ distribution is visible. The distribution of gold inclusions correlated with cyanide but not with silicate and calcium oxide. The TOF-SIMS and mass analyses detected Au and other related, stable complexes. Chemisorption and physisorption mechanisms by carbonaceous matter and silicates could be involved in gold retention as action mechanisms.

**Author Contributions:** Conceptualization, G.J.M.M., F.R.V. and J.E.C.S.; methodology, J.E.C.S., E.M.G.R., L.G.C.A. and G.J.M.M.; formal analysis, J.E.C.S., E.M.G.R., G.J.M.M. and F.R.V.; investigation, J.E.C.S., E.M.G.R., G.J.M.M., Y.K. and F.R.V.; resources F.R.V. and G.J.M.M.; supervision, Y.K. and L.G.C.A.; writing—original draft preparation, J.E.C.S., E.M.G.R., G.J.M.M., Y.K. and G.J.M.M.; writing—review and editing, L.G.C.A., G.J.M.M. and F.R.V.; project administration, F.R.V., G.J.M.M. and J.E.C.S. All authors have read and agreed to the published version of the manuscript.

**Funding:** This work is supported and financed by Cal y Cemento Sur S.A. E. Garcia is grateful to CONAHCYT for supporting his Ph.D. studies at UAdeC (scholarship number 888159).

**Data Availability Statement:** The original contributions presented in the study are included in the article, further inquiries can be directed to the corresponding author.

**Conflicts of Interest:** Authors G.J.M.M. and F.V.R. were employed by the company Cal y Cemento Sur S.A. The remaining authors declare that the research was conducted in the absence of any commercial or financial relationships that could be construed as potential conflicts of interest.

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
