# Peer review of "Study of Preg-Robbing with Quicklime in Gold Cyanide Solutions Analyzed by Time-of-Flight Secondary Ion Mass Spectrometry"

_metals, doi:10.3390/met14040416_

Round 1

Reviewer 1 Report

Comments and Suggestions for Authors

In this paper, the retention of gold in cyanide solution by quicklime has been systematically studied. The research work has certain reference value for the adsorption of metal elements by carbon-based substances and silicates in cyanide solution.

1)     It is mentioned in lines 36 and 37 of the Introduction that the calcination of limestone to obtain quicklime will produce CO2 during the heat treatment process, how much CO2 will be produced and how to control CO2 emission.

2)     Section 2.2, line 147, mentions the use of distilled water to dilute the solution, why distilled water is used, and whether it can be replaced with other water that can be produced industrially.

3)     Is it inappropriate to say in section 3.1, line 249, that oxides and sulfides may have interfered with the results in some way, but the test results did not change much?

4)     t is mentioned in line 261 of Section 3.2 that there is a 5.803% error between the excess ratio and the exact ratio test. Is there any other literature supporting that this error is smaller?

5)     Can SEM in Section 3.6 provide clearer image explanation?

6)     In Figure10, it is suggested to improve the XRD pattern, including transverse and longitudinal coordinates and scales. The current pattern cannot explain the problem.

Reviewer 2 Report

Comments and Suggestions for Authors

The manuscript consists of the analysis of the degree of preg-robbing by quicklime in gold cyanidation solutions, as well as the influence of the mineralogy associated with the quicklime and which causes an alteration of the preg, and can act synergistically to carry out this action.

The manuscript is well written and minor corrections are needed. I provide some suggestions that would help to improve it.

Line 30. Please describe the action

Line 31.  Quicklime contains bicarbonates ????

Line 41, Please, put an example of pre-robbing minerals

Line 98-99. Replace “Quicklime contains both species of carbonaceous matter and silicates” by “Quicklime is associated with both species of carbonaceous matter and silicates”.

Line 106. Materials: qlvk, qlrk,qlem, rk+20%, rk+5%, rk+28%, and rk+35% are seven samples, what about the other two samples?.

Line 122. Table 1 should be cited in the text previously to be shown. Then, change the location of the table to after line 130.

The same problem occurs with Table 2.

How did you analyze the Au contents?

If quicklime materials contain silicates and carbonaceous matter, it should be interesting to show here the chemical composition of samples.

Line 149. XRD, FTIR. 1. The first time an acronym is written, the full name must also be given.

Line 190 The same for SEM-EDS acronyms.

Results

Line 229. You cannot start a paragraph by showing a figure. We must first introduce it and remember that all Figures must be cited in the text before the images appear.

Gold composition and gold losses. be careful with the analytical error, they are using three decimal places. First of all, there is no explanation of how they have analyzed the gold and the accuracy of the method.

Figure 2. Why Y axis reach up to 5.0?, change this up to not more than 2.0 loss %. X asis: write the labels of experiment vertically.

Figure 3 would provide more information if two graphs were made, one for experiments T1 to T10 and the other for experiments T11 to T13.

Figure 4. it would look better if the X axis was up to 2θ=45.

Line 377. The results are shown in Table 3. In this table reduce decimal places from the two last rows.

Figure 5 does not exist.

Figures 4 and 6. The result of samples qlvk and qlrk are exactly the same, then, you do not need to include oth in the graph. Only for the Raman diagram is different. You didn’t comment this difference; it should be interesting if they tried to justify it and if they do not find a way, at least mention it.

Table 4. Phases and silicates present in the samples. Silicates are also phases, then change by “Minerals present in the samples, in weight %”

Figure 8. The Figure caption do not correspond with the graph shown here. The Y axis needs a Title.

Figure 9. The number of microns of the scale bar are blurred and cannot be read. They have some green and yellow letters (???).

Table 5. I consider that one decimal place is enough for all the elements.

Line 618. The same as previously, first explain and later show the images. The font of the name of the species could be significantly lower. I think that you can distribute these images different to observe all in the same page. E.g. you can include four in the same row, and then you only need three rows to show all of them.

Conclusions

Line 700. Delete “plus the DIFRAC.EVA software”. I prefer conclusions without an exhaustive description of what information was provided by each analytical technique.

Comments on the Quality of English Language

In general, I found the English to be correct.

Reviewer 3 Report

Comments and Suggestions for Authors

Study of Preg-Robbing with Quicklime in Gold Cyanide  Solutions, Analyzed by TOF-SIMS is very interesting paper. Some improvement is required!

Line 2: Study of Preg-Robbing with Quicklime in Gold Cyanide  Solutions, Analyzed by TOF-SIMS (please to write full name of TOF-SIMS methods)

Line 39: What is a temperature for thermal decomposition of calcium carbonate fo produce Quicklime

Line 93: They suggest that the tendency to perform preg-robbing is greater in carbonaceous matter than silicates (can you write maximal values of gold  losses)

Line 257: As described above, the tests with different granulometry did not show different results. Why? What is a reason for these results?

Line 433: Water vapor (H2O) reacts with the carbon structure to produce carbon monoxide (CO). Please to add a reaction temperature?

Line 594: For obvious  reasons, special attention was paid to identifying the gold peak (in which form is gold expected?)
